# New Zealand Open Environmental Science Data sets

**Albert Bifet**
University of Waikato

**Jacob Montiel**
University of Waikato

**Nick Lim**
University of Waikato

**Gregory Pearson**
Metservice

**Sebastian Delaux**
Metocean

**James McLeod**
Waikato Regional Council

**Phil Mourot**
Waikato Regional Council

## Abstract

Data Science on environmental spatio-temporal data is becoming a critical and challenging research topic due to the changing nature and rapidly increasing volume of available data. To this end, we would like to introduce TAIAO data repository, comprising of over 30 datasets of various types including images, videos, textual and tabular data.

## 1 Introduction

Environmental data science is strategically essential to New Zealand because it supports, leverages and boosts research on climate change impacts, adaptation and conservation [6]. Effective data science can take an essential role in the Government's goals of improving the quality of freshwater and to reach zero carbon by 2050. Environmental time series or data streams are found in many practical applications in New Zealand. They can consist of monitoring observations or modelling output of flow (e.g. wind, current, water level, ice flow, ice height), concentration (e.g. suspended sediment, nutrients, contaminants), physical properties (e.g. temperature, density) and external forcing (e.g. gravity, solar radiation).

Data Science on environmental spatio-temporal data is becoming a critical and challenging research topic due to the changing nature and rapidly increasing volume of available data. New methods are also required for automatic routine monitoring of biological variables (e.g. birdsong listening stations, automated tree-ring measurements). Environmental time series data need specific processing techniques because:

- **Decisions** (for example, with respect to management regimes or policies) are made over time on the basis of partial information, and we do not have the time to collect perfect and complete datasets;
- the properties of the information are likely to evolve over time (concept drift), violating the assumptions of some standard statistical approaches;
- the information has a history that is difficult to delimit, yet incorporating history can substantially improve predictive power; and
- the information can be multi-scale, ranging from broadscale satellite-derived data to irregularly-spaced point measurements (e.g of temperature, wind velocity, water flow);

Submitted to the 35th Conference on Neural Information Processing Systems (NeurIPS 2021) Track on Datasets and Benchmarks. Do not distribute.

Remotely-sensed environmental data are often taken as archetypes of big data because they exhibit three key properties:

- **Volume:** with multiple constellations of satellites offering daily global coverage at sub-metre resolutions (e.g. over 25 PB of data per day from the ESA Sentinel satellites alone [5]), or geostationary satellites such as Himawari 8, and airborne and terrestrial datasets such as LiDAR and sonar datasets that capture centimetre resolution topographic information over wide areas, the volume of remotely-sensed data is enormous;

- **Variety:** earth observation data are multimodal, comprising observations by both active and passive sensors and across the electro-magnetic spectrum, and packaged in multiple formats (raster/vector, structured/unstructured), but are typically sampled unselectively presenting major challenges for pattern recognition and interpretation; and

- **Velocity:** image cadence has increased dramatically (e.g. the Himawari 8 satellite over NZ produces images every 10 minutes), and growing archives of archival image stacks that are ripe for temporal analysis [4].

## 2 Time-Evolving Data Science / Artificial Intelligence for Advanced Open Environmental Science (TAIAO)

TAIAO[2] is a New Zealand government supported, multi-year, multi-million dollars, programme aimed at improving the data capabilities of researchers in New Zealand. TAIAO was launched in 2020 and is currently in the early stages of the development of the platform. The motivation of the TAIAO programme includes advancing the state-of-the-art in environmental data science by developing new machine learning methods for time series and data streams with the capacity to deal with large quantities of big data in real-time, with special emphasis on processing the data collected on the New Zealand environment. TAIAO also aim to build an open-source framework to implement machine learning on time series data, as well as provide an open available repository with datasets to improve reproducibility in environmental data science. Ultimately, TAIAO aim to democratise and build capability in fundamental and applied data science.

This programme is a multi-institute, multi-domain and includes data scientists, data engineers, environmental scientists, and machine learning researchers from undergraduate to post-graduate level. Moreover, collaboration is expected to extend beyond technical aspects to include regional councils, iwi[1] and co-governance entities to implement the methods we develop to support governance and management decisions with analyses based on large volumes of data that they cannot currently process.

### 2.1 Reproducible Notebooks

TAIAO uses Jupyter Notebooks to document and visualize the codes for better reproducibility and documentation. Each notebook is associated to a task and describe how the data can be accessed. The notebooks also documents the application of the dataset as well as the questions the data provider and researcher seek to answer with the data. As part of TAIAO's commitment to open-source platforms, we use Jupyter notebooks as it is open-sourced, light-weight while being capable. We envision that as more notebooks in the platform are developed, we can improve the transparency and the reproducibility of the findings, as well as improve the accessibility of data science research. Figure 1 describes a conceptual view of the platform and how the corresponding components of the platform interact.

### 2.2 Indigenous Data Science

TAIAO and the New Zealand government is committed to "Vision Mātuaranga" [7] which aims to unlock the potential of traditional indigenous knowledge and recognize the value of the generations

---

[1]largest social units in Aotearoa Māori society.

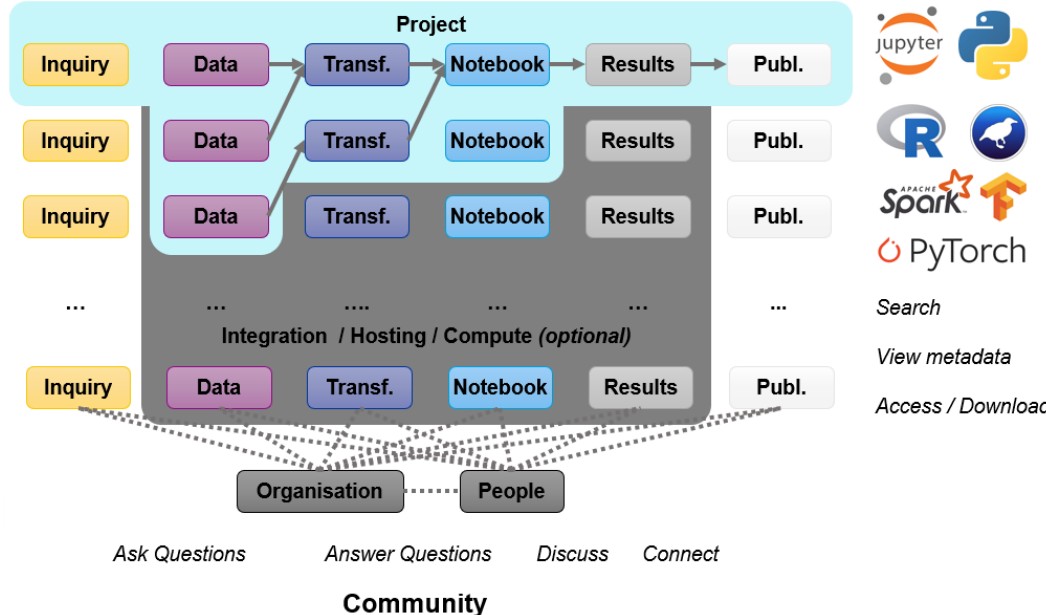

Figure 1: Conceptual view of the platform and the interaction of the corresponding components the TAIAO platform

of tradition and knowledge passed down through the people living in the land. As part of this commitment, TAIAO recognizes the community as partners in science and innovation and guardians of the natural resources and indigenous knowledge.

TAIAO works closely with the local co-governance entities and the iwi to understand the interests and issues of the community. In addition, TAIAO also hold regular dialogues and meetings to review the environmental findings and review the capability developed as well as to democratize the platform and to build the data science capabilities of the community.

In addition to that, TAIAO is also committed to principals of "Te Mana Raruanga" [1] which recognizes Indigenous data rights and sovereignty. Here, TAIAO recognizes the rights and ownership of the data and are have regular dialogue with the owners of the data regarding how the data is being used and the rights and accessibility to the data.

## 3 Available datasets

Over the last year (2020), we have been building a repository of environmental data and hosting the data within the TAIAO platform. Currently, the TAIAO environmental dataset repository currently contains a collection of over 30 datasets of various type including images, videos, textual data, and multi-variate tabular data

Table 1 is a summarises of some of the datasets in the platform. Note that additional datasets are still being added and the list in the table may not include some of the more recent datasets.

Table 1: Summary of the datasets in the TAIAO data repository

| Name | Description | Category/Tag | Type |
|---|---|---|---|
| Aggregated wave and atmospheric forecast derived from GFS guidance | Hindcast and forecast wave and atmosphere model | Model, Forecast, Sea, Wind, Wave | Tabular |
| Continued on next page | | | |

| Name | Description | Category/Tag | Type |
|---|---|---|---|
| Ardmore airport automatic weather station | Nowcast percipitation, temperature and barometeric, updated every 7.5 minutes | Observation, Land, Wind, Temperature | Textual |
| Ashburton Aerodrome automatic weather station | Nowcast percipitation, temperature and barometeric, updated every 7.5 minutes | Observation, Land, Wind, Precipitation, Temperature | Textual |
| Auckland airport automatic weather station | Nowcast percipitation, temperature and barometeric, updated every 7.5 minutes | Observation, Land, Wind, Precipitation, Temperature | Textual |
| Auckland-Hamilton Corridor Aerial Photography | Aerial photograph on the Auckland-Hamilton Corridor region | Observation Image | Image |
| Birchwood automatic weather station | Nowcast percipitation, temperature and barometeric, updated every 7.5 minutes | Observation, Land, Wind, Precipitation, Temperature | Textual |
| Blitzen lightning feed | Archive of lightning records and live feed | Observation, lightning, location, time, intensity, type | Tabular |
| Campbell Island Wave buoy | Nowcast Wave heigth, Wave direction, Wave period | Observation, Sea, Wave, Buoy, Wave heigth, Wave direction, Wave period | Tabular |
| Coromandel river + rain gauge time series | Time Series of rivers and ran gauages in Coromandel Peninsular dated from 2010-2020 at 5 minutes resolution | Observation, time, intensity | Tabular |
| Flat Hills Automatic weather station | Nowcast percipitation, temperature and barometeric, updated every 7.5 minutes | Observation, Land, Wind, Precipitation, Temperature | Textual |
| Global topography (elevation, slope, and aspect) | Topography metrics; ARD tile format. Extracted from the NASA Making Earth System Data Records for Use in Research Environments (MEaSUREs) version of the Shuttle Radar Topography Mission (SRTM) global elevation dataset | Land, topography | Image |
| Google Earth Engine global geospatial data sets | Planetary scale satellite imagery and derived data products | Cimate, weather, geophysical data products, satellite imagery (remote sensing) | Image |
| GPATS Oceania lightning feed | Archive of lightning records and live feed for Oceania area | Observation, lightning, location, time, intensity, type | Tabular |
| | | | Continued on next page |

| Name | Description | Category/Tag | Type |
|------|-------------|--------------|------|
| Haast automatic weather station | Nowcast percipitation, temperature and barometeric, updated every 7.5 minutes | Observation, Land, Wind, Precipitation, Temperature | Textual |
| Himawari-8 2km half disc archive | Archive of himawari-8 satellite data | Observation, Remote-sensing, visible, in-frared | Image |
| Himawari-8 500m channel 3 half disc archive | Archive of full resolution channel 3 data from Himawari-8 satellite | Observation, Remote-sensing, visible | Image |
| Himawari-8 AWS NOAA archive | Full resolution archive of Himawari-8 satellite (data from end of 2019 only) | Observation, Remote-sensing, visible | Image |
| Landsat 8 remote sensing data | Global analysis ready multispectral satellite data from Landsat 8 OLI sensor | Land, remote sensing, multi-spectral | Image |
| LILA Wellington Camera Trap | 270450 Images of wildlife from 187 camera traps locations[3] | Observation Image | Image |
| Moana New Zealand Hydrodynamics Re-analysis v1.9 | Hydrodynamic reanalysis of new zealand waters. | Model, Hind-cast, Sea, Cur-rent, SST | Tabular |
| Mt Karioi predator camera video feed | 2101 videos of wildlife from 20 camera trap locations. Note that this dataset is available only on request and with permission from the Hapu data owners | Observation, Video, Image | Video |
| Mt Karioi predator trap logs | Table of status of traps, including description of bait used, deployment date and date last checked | Observation, Manual logging | Tabular, Textual |
| NZ rain radar archive - RAW data | Archive of data from MetServices doppler radar network. | Observation, Atmosphere, Reflectivity | Image |
| Regional Council Water quality and Discharge data | | Water, rainfall | Tabular |
| Sentinel 1/2 snapshot of waikato region | Hyperspectral satellite image from Sentinel 1/2 | Observation Image | Image |
| Southern Ocean Waverider buoy | Nowcast Wave heigth, Wave direction, Wave period | Observation, Sea, Wave, Buoy, Wave heigth, Wave direction, Wave period | Tabular |
| TOA lightning feed | Archive of lightning records and live feed for Oceania | Observation, lightning, location, time, intensity, type | Tabular |
| Tropical cyclone archive | Archive of tropical cyclone trajectory | Observation, cyclone, trajectories | Tabular |
| | | | Continued on next page |

**Table 1 – continued from previous page**

| Name | Description | Category/Tag | Type |
|---|---|---|---|
| Horizons Air Quality | PM10 and PM100 particulate data taken at 5 minute samples | Air Quality | Tabular |
| Hawke's Bay Air Quality | PM10 and PM100 particulate data taken at 5 minute samples | Air Quality | Tabular |
| Hawke's Bay Air Quality - Raw | PM10 and PM100 particulate data taken at 5 minute samples | Air Quality | Tabular |
| Waikato Region Aerial Photography | Orthorectilinearized projection of aerial photography of the Waikato region taken at 0.03m resolution | Observation Image | Image |
| GFS | Historical percipitation forecast of the Coromandel region from 2015-2018 [8] | Water, Rainfall, Climate | Tabular |

Table 1: Summary of the datasets in the TAIAO data repository

## 3.1 Strategic Use of Big Data Sets

Unlike other centres of data science expertise such as Europe, North America and Asia, New Zealand is an island with a low population density and a high level of urbanisation. Its weather changes quickly and in ways that are difficult to predict, and it has a low density of on-the-ground environmental data measurements (very low in surrounding oceans and on land away from population centres), so it relies heavily on satellite measurements and numerical modelling predictions, and has to combine broad-scale satellite data with sparse on-the-ground data. The TAIAO project particularly focuses on the challenges of making that combination effective.

Moreover, NZ climate is at the interface between tropical and polar air masses, and its coast is connected to the Southern and Pacific Oceans and the Tasman Sea, which is an ocean-warming hotspot. There is a need for fit-for purpose tools that are particularly tailored to this complex environment since existing methods developed overseas are often not suitable for nor transferable to New Zealand conditions.

Topographic datasets are also relatively sparse compared to more populated land-masses; this gap is being corrected by (for example) the national LiDAR survey that is underway, funded by the Provincial Growth Fund and regional councils[2], but integrating the LiDAR data with other spatial data present challenges that the TAIAO project is well-placed to address. To meet these challenges, the TAIAO project will use large-scale datasets facilitated by research partner MetService and build on ongoing work of the environmental scientists within its team. The goal is to build on state-of-the-art modelling datasets of coastal ocean circulation, connectivity and marine temperature being developed in MetService's Moana project[3] and weather radar images, weather station data and high-resolution satellite imagery archived by MetService. It is expected to use existing and newly-acquired LiDAR datasets.

Going beyond physical data to biological data, a joint project between the University of Waikato and Xerra uses estuarine colour indices from Sentinel II satellites to detect estuarine ecosystem tipping points[4].

## 4 Future plans and capabilities

TAIAO is currently in the early stages of development, and over the next few years, we plan to improve the platform to better index the datasets. We also plan to increase the number of notebooks as well as to include more complex, cross-domain, multi-dataset examples showing the application of

---

[2]`https://www.linz.govt.nz/data/linz-data/elevation-data`

[3]`https://www.moanaproject.org/`

[4]`https://www.xerra.nz/2019/06/11/calibrating-satellite-imagery-using-ground-based-data-collection/`

cross-domain datasets. Additionally, we plan to improve the process of adding additional datasets and notebooks to improve the accessibility and contribution from the TAIAO community

## 5 Conclusion

While the TAIAO project is relatively young, we have compiled a repository of varied and unique datasets that are pertinent to environmental research. We are confident that the TAIAO project can improve the accessibility of data science research especially in the field of environmental science

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
