# OpenReview forum: "New Zealand Open Environmental Science Data sets"
_NeurIPS.cc/2021/Track/Datasets_and_Benchmarks/Round1 — Submitted to NeurIPS 2021 Datasets and Benchmarks Track (Round 1)_

### Official Review · Reviewer_Cq6L · 2021-06-19
**An interesting repository, but paper looks thin and out of scope**

**Rating:** 3
**Confidence:** 2
**Clarity:** The paper is clearly written.

**Strengths:**

The datasets are a valuable resource that is openly accessible to the community, and can be a useful asset for time series prediction.

**Weaknesses:**

I do not think this paper is within the scope of the track, as it is not specifically oriented to machine learning or has any concrete machine-learning content. The paper presents the repository of datasets, which of course could be used for machine learning, but it could perfectly be a paper in an environmental science journal. For the NeurIPS dataset track I would expect to see more focused papers that not only present the data, but make some effort to show in what ways it can be useful for the community - in this paper in its current state, the mentions to machine learning are very abstract or referring more to future plans than to concrete work done.

Furthermore, the paper is rather thin on content - more than two pages are taken by a table of the datasets, and much of the text is rather generic. I do not think the reader learns a lot from reading the paper in its current state.

**Additional Feedback:**

No author rebuttal has been posted.

Some typos:

l. 55: This programme is a multi-institute, multi-domain -> missing a noun after "multi-domain".

l. 63: Each notebook is associated to a task and describe -> Each notebook is associated to a task and describes

l. 87: Currently, the TAIAO environmental dataset repository currently -> remove one "currently"

l. 90: Table 1 is a summarises of -> Table 1 is a summary of

p. 4: ran gauages -> rain gauges

l. 127: missing final period.

**Correctness:**

The claims are sound as far as I can see. Since this is a repository with many datasets, there is not a lot of detailed information on each.

**Documentation:**

Since this is a repository with many datasets, the paper does not provide focused documentation on each dataset. The website does provide documentation, although it is not always complete (for example in some datasets, the "License" field is "N/A") and some links to data do not seem to work (e.g. for the "Aggregated wave and atmospheric forecast" I obtain "{"message":"You cannot consume this service"}", and for the "Coromandel River & Rain gauge time series" I get a Google prompt to request access, which makes the data not open).

**Ethics:**

The paper describes how the indigenous communities have been involved in the project. I do not see any ethical concerns with this paper.

**Relation To Prior Work:**

Not much contextualization is done with respect to prior work. It would be useful to have references to machine learning work using this kind of data, as well as other similar existing datasets, shortcomings of existing datasets that these ones aim to solve, etc. As it stands, the bibliography and references to prior work are minimal.

**Summary And Contributions:**

This paper presents a data repository that contains various spatio-temporal datasets with text, tabular data, images and videos related to environmental science. The repository is funded by the government of New Zealand and is in its early stages of development, and the project includes the intention to develop methods and a framework for machine learning. The paper describes the use of Jupyter notebooks as a platform to work with the data, the involvement of indigenous communities, provides a list of the current datasets, some comments on their particularities and plans for future work.

---

### Official Review · Reviewer_WRJQ · 2021-07-01
**Review of "New Zealand Open Environmental Science Data sets"**

**Rating:** 4
**Confidence:** 4

**Strengths:**

1. The data repository is large and diverse.

**Weaknesses:**

1. Although the authors argue about the uniqueness of their data, they do not present any empirical evidence to support their claims. Can it be more concretely established how the distributions of various data specific properties differ. Is the difference statistically significant? Are there any case studies that can be conducted only on this data repository?

2. The paper does not clearly present all details about the datasets listed in Table 1. Specifically, what is the volume of each dataset? What is the time-range of each dataset? Are their interlinkages between the various datasets based on time, space or other common factors?

3. The paper does not present any machine learning case-study that brings out the uniqueness of their repository.

**Additional Feedback:**

I do not have any additional feedback other than what's already provided in the other fields.

**Clarity:**

The presentation can be improved. There are also typos and grammatical errors. As an example, the full stop is missing at the end of the first paragraph in Section 3.

In the second paragraph of the same section, the first sentence "is a summarises of.." is not grammatically correct.

**Correctness:**

In section 3.1, the authors argue about the uniqueness of environmental in New Zealand when compared to North America, Europe, and Asia. I would like the authors to substantiate these claims with actual empirical data. For example, the authors mention that changes in weather is quicker in New Zealand than in the other parts of the world. Can this be shown from historical data.

**Documentation:**

The details on the paper are not sufficient as outlined in the weaknesses section.

**Ethics:**

I do not see any ethical concerns.

**Relation To Prior Work:**

No, the paper can be better contextualized by a more thorough comparison with existing publicly available datasets.

**Summary And Contributions:**

The paper introduces a series of environmental datasets from New Zealand, many of which are spatio-temporal in nature. The data repository includes images, videos, textural and tabular data.

---

### Official Review · Reviewer_q398 · 2021-07-02
**Nice list of environmental datasets but lacks analysis**

**Rating:** 3
**Confidence:** 4

**Strengths:**

Environmental modeling is an important application area, and collecting relevant data is valuable.

**Weaknesses:**

However, the paper essentially just lists the datasets and provides some brief, high-level, and often vague discussions of the context of their collection and use.

**Additional Feedback:**

This paper reads like an early draft of what could become an excellent paper. It implies considerable data, but does no analysis of that data or comparison to similar datasets. At a minimum, the authors should select several of the datasets for deeper exploration. For instance: how large are the datasets? Is the data sufficient to train relevant models? How is the data quality measured? What other similar datasets exist, and why should these datasets be used instead? Environment science is a vital area for ML, and it would be fantastic to see a more complete version of this work.



**Clarity:**

The paper is short and relatively easy to read, though sometimes vague and "press release" like in its discussions.

**Correctness:**

The description of the characteristics of environmental datasets makes sense. Given only a sentence or two about each dataset, it is hard to reach any conclusion about methodological soundness or utility.

**Documentation:**

Given the overall quality level of the paper, individually investigating the 30+ datasets did not seem merited.

**Ethics:**

The data is on environmental phenomena such as precipitation and unlikely to have ownership or privacy concerns. The likely societal impact of this data is positive. Collecting and providing it is admirable.

**Relation To Prior Work:**

The paper lacks a significant discussion of related work.

**Summary And Contributions:**

The paper introduces a set of datasets for environmental phenomena time sequences. It briefly summarize why such datasets are important and the unique challenges they present relative to other datasets. The paper then briefly describes how the datasets are supported by the New Zealand government, how Jupyter notebooks are provided, and how indigenous people are consulted. Each of the 30+ datasets are described by a single row in a table.  The paper discusses unique challenges of New Zealand climate, and conveys that the dataset collection effort is in an early stage.

---

### Decision · Program_Chairs · 2021-07-26

**Decision:**

Reject

**Comment:**

The reviewers unanimously agree on rejection and there was no rebuttal. Therefore, I recommend rejection.